## Rapid Communication

depression; quality of care; psychotherapy; low-income countries; global mental health

**Corresponding author:**
Lily Cooke;
Email: k2481125@kcl.ac.uk

# Development and preliminary inter-rater reliability of the new PROOF tool to measure fidelity of problem-solving therapy for depression delivered by non-specialists in a low-resource African setting

Lily Cooke[1,2] , Tarisai Bere[3], Amelia Stanton[4], Walter Mangezi[3], Steven A. Safren[5], Tsitsi Mawere[3], Lena Skovgaard Andersen[6] , Christina Psaros[7], Samantha M. McKetchnie[7,9], Meghana Vagwala[8], Kia-Chong Chua[1], Conall O'Cleirigh[7,8,9], Aya Mitani[10] and Melanie Abas[1]

[1]Institute of Psychiatry, Psychology and Neuroscience, King's College London, London, UK; [2]Rehabilitation and Human Performance, Icahn School of Medicine at Mount Sinai, New York, New York, USA; [3]College of Health Sciences, University of Zimbabwe, Harare, Zimbabwe; [4]Department of Psychological and Brain Sciences, Boston University, Boston, Massachusetts, USA; [5]Department of Psychology, University of Miami, Miami, Florida, USA; [6]Department of Public Health, University of Copenhagen, Copenhagen, Denmark; [7]Department of Psychiatry, Massachusetts General Hospital, Boston, Massachusetts, USA; [8]Department of Psychiatry, Harvard University, Boston, Massachusetts, USA; [9]Harvard University Center for AIDS Research, Harvard University, Boston, Massachusetts, USA and [10]Dalla Lana School of Public Health, University of Toronto, Toronto, Ontario, Canada

## Abstract

Problem-solving therapy (PST) is a brief psychological intervention often implemented for depression. Currently, there are no tools with well-evidenced reliability to measure PST fidelity. This pilot study aimed to measure the inter-rater reliability and agreement of the **Pro**blem-**So**lving Therapy **F**idelity (PROOF) scale, comprising binary 14-item adherence and an 8-item competence subscales. Transcripts were from the TENDAI trial, a Zimbabwe-based PST intervention for depression and medication adherence. Seven transcripts were each rated by seven specialists, and two transcripts were each rated by two non-specialists. Inter-rater agreement was assessed using percent agreement and inter-rater reliability was assessed using Gwet's $AC_1$. The PROOF subscales demonstrated promising inter-rater agreement among specialists (adherence = 90.4%, competence = 82.5%) and non-specialists (adherence = 92.9%, competence = 68.8%). Inter-rater reliability analyses yielded a Gwet's $AC_1$ of 0.411–0.778 and 0.619–0.959 for adherence and competence among specialists, and 0.529–1.00 for adherence in non-specialists. The PROOF scale has the potential to fill the gap of fidelity tools for PST delivery.

## Impact statement

Despite treatment fidelity measures being a vital prerequisite for assessing the efficacy of any psychotherapeutic modality, no problem-solving therapy (PST)-centred fidelity tools have measured inter-rater reliability to date. With the WHO call for the expansion of brief psychological interventions, the demand for this fidelity gap to be met is clear. This study provides strong evidence for the inter-rater reliability of the **Pro**blem-**So**lving Therapy **F**idelity (PROOF) scale as a measure for evaluating PST fidelity. Principally, however, the development of the PROOF scale has provided valuable insights into the design aspects that are most likely to result in a successful and useable tool, and the pitfalls that may limit it. The method of piloting and iterating on item language showed that ambiguity, even with deliberate attempts to minimise, was likely a sustained influence on disagreement. Thus, efforts to maximise objectivity and minimise subjectivity in item language should be viewed as of paramount importance. With further psychometric research, the PROOF scale has high potential for valued contributions in the field of Global Mental Health, as it fills a long-unaddressed gap in PST fidelity measurements.

## Introduction

Brief psychological interventions, typically consisting of 2–10 sessions serviced from evidence-based therapies, are growing in popularity due to the increasing burden of mental health

disorders globally (Roberts et al., 2021). Task-shifting, wherein non-specialist healthcare workers (NSHWs) deliver therapies under the supervision of trained professionals, renders these interventions particularly relevant to low-resource settings, which lack adequate availability of mental health specialists (Chibanda et al., 2016). Treatment fidelity can be defined as the degree to which treatments are implemented as intended, and involves both adherence, the degree to which pre-specified interventions are used, and competence, the skill with which the intervention is delivered (e.g. empathy, use of open questions) (Fonagy and Luyten, 2019). When task-shifting, it is vital to ensure that NSHWs can deliver with fidelity to permit the safe expansion of NSHW responsibilities (Fonagy and Luyten, 2019).

Problem-solving therapy (PST), a brief intervention, has been shown to have greater effect on depression outcomes than usual primary care ($d = -0.021$, 95% CI = $-0.37$ to $-0.05$, $p = 0.047$), by training people with depression to generate workable solutions to stressful life difficulties (Cape et al., 2010). TENDAI, delivered by NSHWs in Zimbabwe, is a 6-session PST intervention for depression and antiretroviral therapy adherence, alongside simplified behavioural activation, sleep hygiene and stress management (Abas et al., 2022; Nyamayaro et al., 2020). Despite the critical role of assessing NSHW fidelity in delivering PST, few reliable tools exist. Current tools, such as PST-PAC, ENACT and EQUIP are appropriate for assessment of role play and/or training of their respective interventions; however, lack the dual assessment of both

adherence and competence warranted for our setting and were not designed to rate audio recordings of sessions (Hegel et al., 2004; Kohrt et al., 2015; Pedersen et al., 2021). This study, nested within the TENDAI intervention, set out to address this gap through the development and assessment of a PST adherence and competence tool.

## Methods

### PROOF development

The **Pro**blem-**So**lving Therapy **F**idelity (PROOF) scale includes both an interventionist adherence and competence subscale. The adherence subscale measures the interventionist's integrity in delivering each element of the TENDAI intervention as protocolised. Details of the TENDAI intervention design and session components can be found in Figure 1 and Table 1. The last author, a UK-based psychiatrist (MA), co-designed the subscale with a Zimbabwean clinical psychologist (TB) and two US-based clinical psychologists with experience working in Zimbabwe (CO, AS), using the TENDAI interventionist manual and PST-PAC. The adherence items were tailored to session-specific content, and a binary rating system was employed.

The competence subscale evaluates the interpersonal skills of the interventionist and PST-relevant therapeutic factors. The competence subscale is consistent across all TENDAI sessions and was

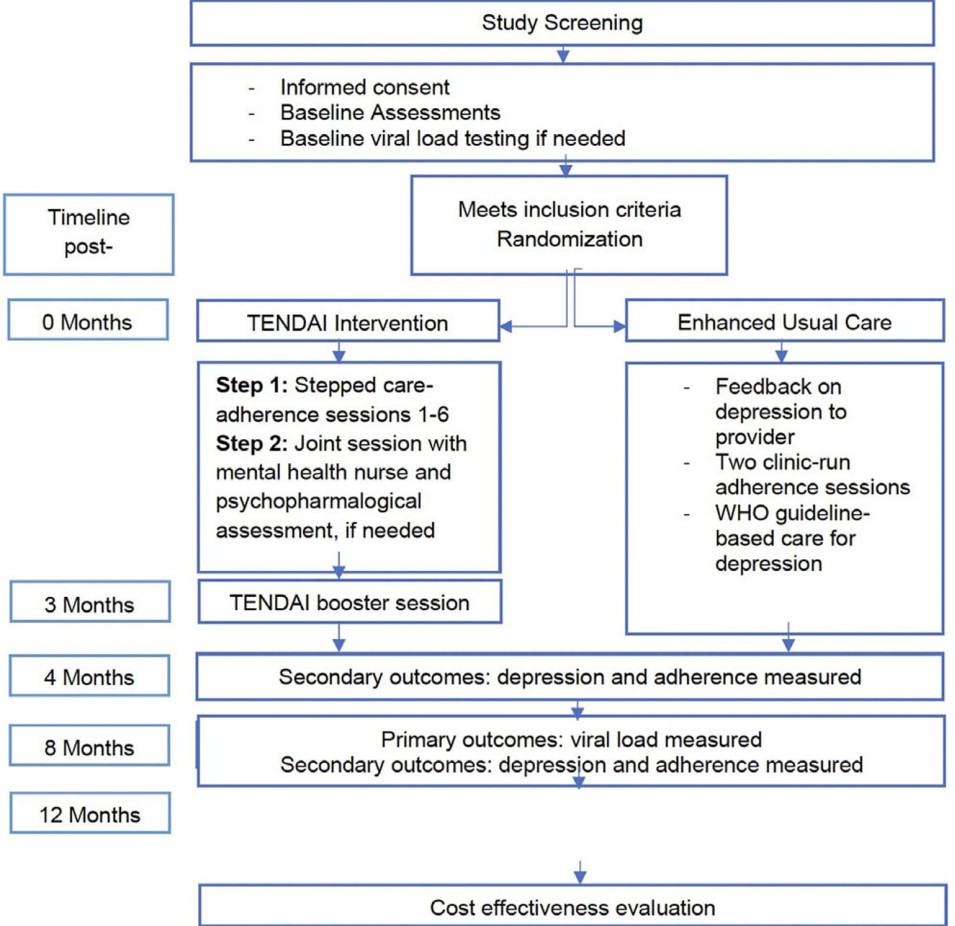

**Figure 1.** TENDAI and enhanced-usual-care intervention design.

**Table 1.** Summary of six TENDAI sessions and booster session

| Session number and theoretical basis | Session-specific content |
|---|---|
| Session 1 – life steps | Focuses on antiretroviral adherence through a locally adapted version of the adherence intervention, *Nzira Itsva*. Motivational interviewing, goal setting, problem solving, and video-delivered education provide a solid ground for addressing the barriers to adherence. By highlighting participant's life goals and tying them to optimal antiretroviral adherence, this session is followed up using adherence booster sessions throughout the trial. |
| Session 2 – problem-solving therapy | Centred around psychoeducation of depression and problem-solving therapy. The cultural idiom of 'thinking too much' and the vicious cycle of thinking are incorporated into the psychoeducation through storytelling. A specific problem is collaboratively identified and worked through, *via* identifying potential solutions. A solution is agreed on for homework. |
| Session 3 – problem-solving therapy | Psychoeducation and problem-solving strategies are reviewed. The previous sessions homework and any potential barriers in achieving it are reviewed, and further solution strategies are identified. The collaboratively agreed on solution is established for homework. |
| Session 4 – problem-solving therapy | Psychoeducation, solution development strategies, and the previous week's homework is reviewed. Interventionists then guide the patients to complete an adaptive activity for the following weeks homework. For example, activities may include a physical activity, a social activity, or an activity that promotes a sense of achievement. |
| Session 5 – problem-solving therapy | A check-in on antiretroviral adherence and the previous week's homework is completed by the interventionist. Further psychoeducation around depression is administered, focusing on skills for good sleep and relaxation. Homework is collaboratively established to be completed before the final session. |
| Session 6 – problem-solving therapy | Reviews depression psychoeducation and problem-solving strategies for antiretroviral adherence and life problems. A relapse prevention plan is formed, focusing on depression relapse triggers, warning signs, coping strategies, and self-care activities. |
| Booster session | Occurs 6 weeks following the final session and reviews depressive symptoms, antiretroviral adherence, and when relevant, depression medication adherence. |

adapted from ENACT (Kohrt et al., 2015). Each ENACT item was analysed for relevancy in evaluating PST fidelity and culturally adapted through the rewording of certain items to include culturally specific terms and idioms, originating from qualitative work surrounding the Zimbabwe-based Friendship Bench and TENDAI trial (Abas et al., 2016; Chibanda et al., 2017). Competence items were initially scored as '0' (not demonstrated), '1' (partially demonstrated) and '2' (demonstrated well). During the pre-pilot phase, the competence items were revised to a binary rating scheme of '0' (not demonstrated) and '1' (demonstrated) to increase the feasibility of use in the local context.

## Rater selection

The rater team included seven mental health specialists (six practicing clinical psychologists and one psychiatrist), and two non-specialist general nurses. The non-specialists worked on the TENDAI trial as research assistants and had been trained in the components of the PST intervention. Four of the specialists had trained the interventionists for the TENDAI trial. Specialist raters had at least four years of experience in training of, and/or research on, psychological therapies for depression in people living with HIV in low resource settings, with five out of seven having more than 10 years of experience.

## Study design

This pilot study focuses solely on Session 2 of TENDAI as it is the first to introduce PST and psychoeducation for depression, whereas Session 1's focus is primarily antiretroviral therapy adherence. A fully crossed design was chosen to assess inter-rater reliability, whereby all transcripts were assessed by all raters (Hallgren, 2012).

Initially, a pre-pilot 'mock trial' study was conducted where three of the specialist raters and two of the nonspecialist raters each rated 12 sessions of therapy for the purpose of discussion aimed at language refinement of scale items and reproducibility optimisation. The raters were members of the research team, and the scoring sheets were completed and stored on REDCap (Harris et al., 2009, 2019). Subsequently, two further mock trials were conducted using the refined subscales on only Session 2. These subsequent trials involved the complete team of seven specialists and two non-specialist raters to familiarise all raters with the rating process and to resolve any outstanding item ambiguity. The mock trials led to revisions of the tool and resulted in a final 14-item adherence subscale and 8-item competence subscale (Supplementary Tables S1 and S2).

With the finalised tool, specialist raters ($N = 7$) rated the same seven transcripts, while the non-specialist raters ($N = 2$) assessed two of these seven transcripts. All ratings were recorded and stored on REDCap. Transcripts for both the mock and true ratings were randomly selected, translated by a local clinical psychologist who was a native language speaker, and anonymised. Interventionist codes were assigned to ensure that raters were unaware of who was delivering the sessions. Ethical approval was granted by the Institutional Review Board of King's College London (LRU/DP-21/22–29,822).

## Data analysis

Percent agreement, the percent of ratings that agree, was originally used to measure inter-rater reliability, that is, the consistency between raters. This application has since been discredited due to its inability to account for chance agreement and the measure is now used solely to assess inter-rater agreement, with researchers having turned to reporting both inter-rater agreement and reliability (McHugh, 2012; Mitani et al., 2017). Gwet's $AC_1$, an inter-rater reliability statistic, is commonly used for data whose ratings have an unbalanced distribution (Eugenio and Glass, 2004; Wongpakaran et al., 2013). Due to the nature of fidelity tools, Gwet's $AC_1$ was believed to be the most appropriate reliability metric.

Inter-rater reliability and agreement were calculated for each PROOF adherence and competence item. This by-item analysis was chosen to aid in further edits of the tool. The specialist and non-

specialist data were analysed separately. Percent agreement was used to assess inter-rater agreement for both rater teams and subscales. Due to the low availability of non-specialist raters and transcripts, the two mock trial transcripts were included in the adherence dataset for non-specialists. Adherence and competence inter-rater reliability for both rater teams was assessed using Gwet's $AC_1$ (Wongpakaran et al., 2013). The small sample size and significant revisions made during the pilot prevented a formal analysis of non-specialist competence, though percent agreement was reported.

## Results

### Inter-rater reliability and agreement of ratings by specialists

The specialist-rated adherence subscale yielded an average percent agreement of 90.4%. Of the 14 items, 6 showed complete agreement (100%) and 10 showed greater than 90% agreement. Inter-rater reliability analysis ranged from 0.411 to 0.778, with items 8 and 9 showing moderate agreement and items 1–7 and 10–14 reporting substantial agreement.

Inter-rater agreement analysis of the specialist-rated competence subscale showed a range of 61.9–95.9% agreement, averaging 82.5%. Inter-rater reliability analysis using Gwet's $AC_1$ ranged from 0.619 to 0.959, with items 1, 2, 4 and 5 representing substantial agreement. Excellent agreement was found for items 3, 6, 7 and 8.

All results can be seen in Table 2.

### Inter-rater reliability and agreement of ratings by non-specialists

The NSHW ratings resulted in an average percent agreement of 92.9% for adherence. By-item analysis ranged from 75 to 100% agreement. Complete agreement occurred in 10/14 items. Inter-rater reliability analysis produced a Gwet's $AC_1$ range from 0.529 to 1.

Due to the small non-specialist sample for competence, only inter-rater agreement analysis was attainable. Percent agreement ranged from 0 to 100%, with perfect agreement (100%) occurring in 4/8 items. The average percent agreement was 68.8%. For full data on non-specialist ratings, see Supplementary Table S3.

## Discussion

We found high inter-rater reliability for the adherence items among specialist raters (>90% overall agreement). Perfect agreement was observed on key PST elements, such as psychoeducation (items 5 and 6), brainstorming solutions (item 10) and discussing pros and cons of solutions (item 11). Given these strong results, the applicability of the PROOF scale to other PST interventions is promising. Compared to PST-PAC, PROOF had notably improved relative disagreement rates: 31.4% lower for specialists and 49.3% lower for non-specialists (absolute percentage agreement PST-PAC = 86%, PROOF = 90.4% and 92.9%) (Hegel et al., 2004). Interestingly, we found that the agreement between non-specialists was similar to the agreement between specialists. While NSHWs are not typically recommended to conduct treatment fidelity assessments, these initial signs of high inter-rater agreement suggests that their roles could be expanded to include intervention evaluation should an effective tool be validated.

The results of the competence subscale were less definitive with the specialist ratings resulting in an overall percent agreement of 82.5%. The weaker agreement results may have been a consequence of the relatively subjective nature of interventionist competency compared to adherence, or ambiguous item language. Items thematically aligned to reviewing and collaboratively establishing homework (items 3 and 7), providing praise and encouragement (item 6) and delivering clear and non-stigmatising psychoeducation (item 8) resulted in the strongest Gwet's $AC_1$ values representing 'excellent agreement'. Beyond the specialist raters, the non-specialist competence results were limited to an analysis of inter-rater agreement owing to the low availability of non-specialist raters. Our pre-pilot work led to us adding the competence item of 'giving praise' to the PROOF scale as this was considered culturally important in Zimbabwe but was not present in ENACT.

Despite these positive results, there were several limitations to this study. First, the limited, fixed transcript sample size, compounded by low availability of non-specialist raters, may have affected results. These sample size and rater availability limitations resulted in solely inter-rater agreement being reported for the non-specialist rated competence subscale. This was partially mitigated for the adherence subscale through the inclusion of the mock transcript data. However, the competence subscale was heavily edited during the pilot phase, causing the mock trial data to be incompatible for data analysis. Additionally, this pilot study only included one session of the TENDAI trial, thereby providing limited information regarding the scale's transferability to other PST sessions.

The development and assessment of the PROOF scale addresses the lack of PST fidelity tools. For a scalable psychological intervention to be successful, ensuring fidelity must not be a time-consuming and resource-intensive process, given the other resource pressures associated with large-scale implementation. With finite expert supervisors and access limitations, the PROOF tool may provide a feasible and efficient route to quality assurance. To the best of knowledge of the researchers, this is the first PST-related fidelity tool to report both inter-rater reliability and agreement. The study design, based on the first PST-focused TENDAI session, allows transferability to other PST interventions, especially those involving psychoeducation, brainstorming solutions and discussing pros and cons of solutions – all common components of PST interventions. A further strength of this study is its co-production, ensuring cultural relevance. Our immediate next step is to show inter-rater reliability for fidelity to delivering all the remaining elements of the TENDAI intervention. In addition to PST, we will assess inter-rater reliability of fidelity to delivering motivational interviewing, collaborative problem-solving around barriers to taking long-term medication, positive activity scheduling, sleep hygiene and relaxation and relapse prevention. Beyond this, we plan to develop a generalisable, replicable measure that addresses core competencies for components of brief interventions being typically used in low-resource settings, which would apply across multiple interventions.

## Conclusion

The PROOF scale addresses the dearth of PST-centric fidelity tools, with the adherence and competence components demonstrating sufficient performance in their current form. Moreover, its strong inter-rater reliability in non-specialists supports further work

**Table 2.** Inter-rater reliability and agreement of adherence and competence specialist ratings

| | | | | | | Adherence subscale Specialist ratings | | | | | |
|---|---|---|---|---|---|---|---|---|---|---|---|
| Item # | Mean | % Agreement | Gwet's $AC_1$ | 95% CI | *P*-value | Item # | Mean | % Agreement | Gwet's $AC_1$ | 95% CI | *P*-value |
| **1**- Sets agenda | 1.00 | 100% | 0.778 | 0.685–0.872 | **<0.001** | **8**- Identifies problems | 0.63 | 56.5% | 0.411 | 0.257–0.566 | **<0.001** |
| **2**- Reviews medication adherence | 0.94 | 89.1% | 0.691 | 0.498–0.885 | **<0.001** | **9**- Problem defined and specific | 0.82 | 67.3% | 0.504 | 0.338–0.672 | **<0.001** |
| **3**- Reviews homework | 0.89 | 79.6% | 0.608 | 0.491–0.725 | **<0.001** | **10**- Brainstorms solutions | 1.00 | 100% | 0.778 | 0.685–0.872 | **<0.001** |
| **4**- Reviews mood | 1.00 | 100% | 0.778 | 0.685–0.872 | **<0.001** | **11**- Discusses solution's pros and cons | 1.00 | 100% | 0.778 | 0.685–0.872 | **<0.001** |
| **5**- Depression psychoeducation | 0.98 | 95.9% | 0.747 | 0.614–0.879 | **<0.001** | **12**- Agrees on solution | 0.96 | 93.2% | 0.724 | 0.547–0.902 | **<0.001** |
| **6**- Negative cycle psychoeducation | 1.00 | 100% | 0.778 | 0.685–0.872 | **<0.001** | **13**- Sets action plan | 0.96 | 91.8% | 0.708 | 0.628–0.789 | **<0.001** |
| **7**- Problem solving psychoeducation | 1.00 | 100% | 0.778 | 0.685–0.872 | **<0.001** | **14**- Schedules next session | 0.94 | 91.8% | 0.697 | 0.586–0.807 | **<0.001** |
| | | | | | | Mean percent agreement | | | | 90.4% | |

| | | | | | | Competence subscale Specialist ratings | | | | | |
|---|---|---|---|---|---|---|---|---|---|---|---|
| Item # | Mean | % Agreement | Gwet's $AC_1$ | 95% CI | *P*-value | Item # | Mean | % Agreement | Gwet's $AC_1$ | 95% CI | *P*-value |
| **1**- Rapport and empathy | 0.88 | 76.87% | 0.769 | 0.451–0.960 | **<0.001** | **5**- Instils realistic hope | 0.92 | 80.95% | 0.809 | 0.506–1.00 | **<0.001** |
| **2**- Open-ended questions | 0.86 | 76.87% | 0.769 | 0.341–1.00 | **<0.005** | **6**- Praise and encouragement | 0.96 | 91.84% | 0.918 | 0.76–1.00 | **<0.001** |
| **3**- Summarise and reflect | 0.92 | 83.67% | 0.837 | 0.615–1.00 | **<0.001** | **7**- Collaboratively establishes homework | 0.98 | 95.92% | 0.959 | 0.849–1.00 | **<0.001** |
| **4**- Linking of problems | 0.76 | 61.90% | 0.619 | −0.048–0.838 | 0.072 | **8**- Clear and non-stigmatising psychoeducation | 0.96 | 91.84% | 0.918 | 0.760–1.00 | **<0.001** |
| | | | | | | Mean percent agreement | | | | 82.5% | |

*Notes*: Means, percent agreements, and Gwet's AC1s of the PROOF adherence and competence subscale items rated by specialists. CI: confidence interval.

towards deeper integration of NSHWs in intervention evaluation. Future research on the PROOF scale will assess adherence and competence across the whole 6-session intervention. The PROOF scale has high potential to fill an unaddressed gap in PST fidelity measurements.

**Open peer review.** To view the open peer review materials for this article, please visit http://doi.org/10.1017/gmh.2025.10034.

**Supplementary material.** The supplementary material for this article can be found at http://doi.org/10.1017/gmh.2025.10034.

**Data availability statement.** The data that supports the findings of this study is available upon reasonable request from the corresponding author.

**Acknowledgements.** The authors would like to acknowledge the time and work provided by both rating teams.

**Author contribution.** Conceptualization: L.C., T.B., A.S., C.O., M.V., M.A.; Methodology: L.C., M.A. A.M., S.M., M.V.; Data collection: L.C.; Formal analysis: L.C., K.C., A.M.; Supervision: M.A., C.O., W.M.; Writing - original draft preparation: L.C., M.A.; Writing – reviewing and editing: all authors. All authors have read and agreed to the published version of the manuscript.

**Financial support.** The TENDAI trial was supported by the National Institute of Mental Health, NIH (Grant number 1R01MH114708).

**Competing interests.** The authors report no conflicts of interest.

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
