## [Reviewer Report]

Thank you for the invitation to review this interesting manuscript describing the development and preliminary reliability of a new tool to assess fidelity of problem-solving therapy for depression delivered by non-specialists in low resource African setting. The study targets an important topic of both task-sharing as well as ensuring high quality of services delivered. The methodology appear sound, and the findings are encouraging, adding to the impact in the field of global mental health. See few suggestions for consideration:

1. While the authors have provided background on the source of transcripts used for the current study, having some more information in the current paper on the components of intervention would be more useful to understand the scale described later

2. Additionally, can a link or table of components for the TENDAI interventionist manual referenced on page 6.

3. While the raters all have their individual qualifications, if the authors add a line about their extent of involvement in the TENDAI trial or the extent of experience of working in PST interventions, it would be a much clearer read (page 6).

4. On page 7, a mock trial is referred to, the aim of which was to familiarize raters with the rating process. It is not clear as to who conducted the mock trial and how the feedback for the trials was collected .

5. While the authors acknowledge the work done by the ENACT team in designing competence measure for non-specialists, they have also referred to their PROOF scale as filling a long sought-after gap in PST fidelity measures. However, the authors would benefit from referencing the work done by EQUIP team in developing a measure to assess competencies in PST interventions through medium of role plays (https://journals.lww.com/invn/fulltext/2021/19010/development_of_a_tool_to_assess_competencies_of.14.aspx, https://www.cambridge.org/core/journals/bjpsych-open/article/perspectives-on-competencybased-feedback-for-training-nonspecialists-to-deliver-psychological-interventions-multisite-qualitative-study-of-the-equip-competencybased-approach/EE4201B24BE4ED7FE50E66725090E409, https://www.thelancet.com/journals/lanpsy/article/PIIS2215-0366(24)00183-4/abstract )

---

## [Reviewer Report]

This is a useful paper demonstrating the utility and reliability of a measure of fidelity of the delivery of Problem Solving Therapy (PST). It establishes the psychometric properties of the PROOF measure for a particular implementation protocol and context, and demonstrates an appropriate methodology for documenting such properties in PST-related treatment elsewhere.

The Introduction establishes the case for the importance of measuring the fidelity of implementation of scalable psychological interventions and the Method then appropriately documents the process by which this was tackled in this instance. I think some information on the process of producing and analysing transcripts of sessions needs to be presented (given the loss of information that would result from this process compared to, for instance, use of video recordings). In the data analysis sub-section, I think readers would benefit from a clearer elaboration of the distinction and purpose of the respective indicators inter-rater agreement and inter-rater reliability (and the specification in that section of ‘by-item analysis’ - was not all analysis by item?).

The Results are presented concisely and in line with the documented methodology. Given that the content/focus of items is of obvious interest to the reader, I think some information on item focus is relevant to be included in Table 1 (rather than being consigned to the Supplementary Material). Reference is made to broad focus of items in the Discussion, so I think a column indicating in a single word the focus of the numbered item would be both appropriate and feasible.

The Discussion is rather limited in scope and links back to only one published paper. I think a broader elaboration of issues in the quality assurance of scalable psychological interventions is warranted here. In particular, I see a major constraint of the current study to be its focus on a single session of a manualised PST protocol in a single context. It is not unreasonable to expect higher levels of reliability in judgements of interventionist behaviour when the scope of the assessment is so narrow. Is the presumption that researchers (and service quality assurers) should develop bespoke fidelity assessments for each session of a manualised intervention in each intervention context? Or is the goal a more generalisable, replicable measure that addresses core PST competences as demonstrated across multiple intervention sessions (and, potentially, across multiple settings)? I consider that this is an important question that the ‘forward plan’ for research touches on, but does not really address in manner that stands to shape debate and practice.

The manuscript would benefit also from correction of a few grammatical lapses/expression issues including:

Impact Statement: l. 8: ‘their [sic] inter-rater reliability to date’ - tools don’t measure their own reliability‘ line 11: ’This study has provided‘ - better to say ’provides;?; l. 24: ‘ long sought-after gap in PST fidelity measurements’ - I don’t think gaps are long sought after!

Abstract: l. 30: ‘brief psychological intervention often for depression’ - missing word; l. 34: ‘This pilot study aims to’ - better ‘aimed to’?; l.49: better to avoid jargon in this concluding sentence about the ‘tool gap’: ‘The PROOF scale has the potential to fill the fidelity tool gap within PST delivery.’

Intro. l. 5-9: Thonk these opening sentences need unpacking slightly with one or two relevant citations given; l. 51-56: ‘Adherence and Competence Scale (PST-PAC), have not reported reliability or focus exclusively on specific PST interventions or specific settings, such as primary care (Hegel et al. 2004)’ - this sentence is confusing given it references many diverse elements; ‘The demand for this PST fidelity gap to be met is undeniable, as the WHO has called for the expansion of brief psychological interventions (World Health Organization 2016)’ - sadly many facts are now denied so I am not sure the phrase ‘is undeniable’ is the best choice of words'.

Results. l. 48: ‘Inter-Rater Reliability and Agreement for Specialist Ratings’ - isn’t it ‘of Specialist Ratings’?, or perhaps even ‘of ratings by specialists’?? [and similar for non-specialist sub-head]; l. 51: ‘%. By-item inter-rater agreement analysis showed a 56.5-100% agreement range with 6/14 items representing complete agreement (100%) and 10/14 items having >90% agreement’ - I think this can be more clearly worded by simply noting something of the form ‘....of the 14 items 6 showed complete agreement (100%) and 10 greater than 90% agreement’.

Discussion. l. 51 ‘the slightly weaker agreement’ - I don’t think a scientific paper benefits from the addition of slightly here.

Conclusion. l. 44: ‘ demonstrating sufficient performance in their current states’ - I found ‘states’ confusing - ‘current form’?

---

## [Reviewer Report]

Abstract: Statement of specialists and non-specialists reading needs clarification - what does 'same seven transcripts mean in relation to same two transcripts?

Page 6: Lines 3-4: Description of the process is unclear - Is the first session of the trial to introduce PST and psychoeducation of depression?

Page 6: Lines 5-8: Suggest rewrite sentence avoiding split infinitives - A fully crossed design...was chosen.. scale..., whereby all transcripts were assessed by all raters.

Page 6: Lines 9-11 Unclear what is meant by iterative trials...? This description does not appear to be about trials in a formal sense and likely refers to “sessions”. Who delivered these sessions in the pre-pilot study? Were these individuals trained in PST?

Page 6: Lines 14-15 - the expression suggests that some of the seven specialists were involved in the pre-pilot work - this should be clarified as agreement about items then is not completely independent! This is not an issue as Gwet’s AC1 was used in place of Cohen’s Kappa.

Page 6: Lines 22-23: Is there a rationale that explains why non-specialists rated only two of the seven transcripts? How does this impact on the claim that the tool is for use by non-specialists.

General Comment:

The MS provides an interesting account of the PROOF tool around inter-rater reliability. The title is misleading and should perhaps indicate that it is focused on inter-rater reliability.

The MS would also benefit from better describing the context of specialists and non-specialists in which the tool was tested, including the parent study cohort.

PROOF Tool:

While binary scoring assists in reducing complex scoring, it also may mask the nuances that are contained in the Competence Subscale. All of these items are scored as 0, 1, or 2. It is conceivable that a person could straddle 2 or more of these domains! Could the authors comment if the binary approach inflates the level of agreement achieved by simple fact that a 0-2 range has been reduced to Yes/No?

---

## [Reviewer Report]

Thank you for the opportunity to re-review this manuscript, which is well written. I have only one observation in the supplementary material, for the competence scale the authors decided to use a binary approach: “Competence items were initially scored as ‘0’ (not demonstrated), ‘1’ (partially demonstrated), and ‘2’ (demonstrated well). During the pre-pilot phase, the competence items were revised to a binary rating scheme of ‘0’ (not demonstrated) and ‘1’ (demonstrated) to increase feasibility of use in the local context”. But this should be reflected it Table 2 in the left column (0 and 1), to align with the options on the right-hand column (done and not done).

---

## [Editor Report]

Dear Professor Abas

Regarding your paper “Development and preliminary reliability of the new PROOF tool to measure fidelity of problem-solving therapy for depression delivered by non-specialists in a low-resource African setting”, we have received feedback from the peer review process, and there was consensus that the manuscript requires minor revisions. Kindly address all revisions, we look forward to receiving a revised paper.

---

## [Reviewer Report]

I have reviewed the changes made in the manuscript in response to reviewers comments and am pleased to recommen acceptance of the revision.